# Insights into the formation and evolution of extraterrestrial amino acids from the asteroid Ryugu

Christian Potiszil [1] ✉, Tsutomu Ota [1], Masahiro Yamanaka [1], Chie Sakaguchi [1], Katsura Kobayashi [1], Ryoji Tanaka[1], Tak Kunihiro[1], Hiroshi Kitagawa[1], Masanao Abe[2,3], Akiko Miyazaki [2], Aiko Nakato[2], Satoru Nakazawa [2], Masahiro Nishimura[2], Tatsuaki Okada [2,4], Takanao Saiki[2], Satoshi Tanaka[2,3,5], Fuyuto Terui[2,6], Yuichi Tsuda[2,3], Tomohiro Usui [2], Sei-ichiro Watanabe [7], Toru Yada [2], Kasumi Yogata [2], Makoto Yoshikawa[2,3] & Eizo Nakamura [1]

All life on Earth contains amino acids and carbonaceous chondrite meteorites have been suggested as their source at the origin of life on Earth. While many meteoritic amino acids are considered indigenous, deciphering the extent of terrestrial contamination remains an issue. The Ryugu asteroid fragments (JAXA Hayabusa2 mission), represent the most uncontaminated primitive extraterrestrial material available. Here, the concentrations of amino acids from two particles from different touch-down sites (TD1 and TD2) are reported. The concentrations show that N,N-dimethylglycine (DMG) is the most abundant amino acid in the TD1 particle, but below detection limit in the other. The TD1 particle mineral components indicate it experienced more aqueous alteration. Further-more, the relationships between the amino acids and the geochemistry suggest that DMG formed on the Ryugu progenitor body during aqueous alteration. The findings highlight the importance of aqueous chemistry for defining the ultimate concentrations of amino acids in primitive extraterrestrial samples.

The origin of life on Earth is a contentious issue, with a variety of potential environments being proposed, including oceanic hydro-thermal vents[1] and terrestrial hot springs[2]. However, one thing that is certain, is that life would have required amino acids at some point in order to synthesise proteins, which are responsible for a number of biological functions, such as catalysing metabolic reactions, the repli-cation of DNA, transportation of molecules and giving structure to cells and organisms[3]. While it may have been possible to generate amino acids on the early Earth, only extraterrestrial sources have been found to contain abiotically synthesised amino acids with enantiomeric excesses of L-amino acids[4,5]. As such, an extraterrestrial origin for at least some of the building blocks of life has been proposed[6].

An array of simple (low molecular weight) amino acids, such as glycine, alanine, β-alanine, sarcosine and serine, and more complex (higher molecular weight) amino acids have been produced in irra-diated ice experiments[7–9]. Such experiments aim to recreate the

[1]Pheasant Memorial Laboratory, Institute for Planetary Materials, Okayama University, Yamada 827, Misasa, Tottori 682-0193, Japan. [2]Institute of Space and Astronautical Science, Japan Aerospace Exploration Agency, Sagamihara, Japan. [3]The Graduate University for Advanced Studies (SOKENDAI), Hayama, Japan. [4]University of Tokyo, Tokyo, Japan. [5]University of Tokyo, Kashiwa, Japan. [6]Kanagawa Institute of Technology, Atsugi, Japan. [7]Nagoya University, Nagoya, Japan. ✉e-mail: cpotiszil@okayama-u.ac.jp

conditions present in either the interstellar medium (ISM) prior to the start of the solar system, or those present in the protosolar nebula (PSN) shortly after its formation[10]. Therefore, some of the amino acid inventory within carbonaceous chondrite meteorites has been proposed to originate from the ISM or PSN[7,8].

Upon formation of planetesimals in the protoplanetary disk, amino acid bearing ices could have been accreted into such bodies along with other organic compounds and inorganic materials, such as dust grains[4,7,11]. Various isotopic data can be used to try and constrain the region where a given asteroid or parent body accreted. In particular, the $\varepsilon^{48}Ca$ and $\varepsilon^{54}Cr$ values of chondritic meteorites have been useful in understanding the thermal processing history of material accreted by the parent bodies of such meteorites[12,13]. In the case of Ryugu, it was found that some particles demonstrated similar values to CI chondrites, but some plotted towards less thermally processed values[14,15]. Accordingly, Ryugu should have accreted in a similar place within the outer solar system or further out than the CI parent body[14]. Such a finding is interesting, because a cometary origin has been proposed previously for Orgueil[16–18] and more recently for Ryugu[14,19,20]. In such a scenario, the progenitor planetesimal of Ryugu would be expected to accrete large quantities of icy volatiles and thus ISM/outer PSN organic matter, such as amino acids.

While a number of more complex amino acids have been produced in irradiated ice experiments, these are not produced in carbonaceous chondrite like quantities, tending to be at much lower abundances, when compared to more simple amino acids like glycine[7–9]. More complex meteoritic amino acids are thought to require aqueous processing and are thus thought to form on planetesimals while they are hot enough to sustain liquid water[14]. The process of liquid water interacting with a planetesimal to alter its mineral and organic components is termed aqueous alteration. Aqueous alteration is observed to have affected different carbonaceous chondrite meteorites and asteroidal material to different extents, with the most aqueously altered examples being termed petrographic type 1 (e.g. CI1s, Ryugu and CM1s) and the least altered type 3 (e.g. CR3s), with the type 2 s falling in between (e.g. CM2s and CR2s)[21,22]. It is thought that the original accreted crystalline or amorphous silicate minerals, such as pyroxene, olivine and/or amorphous glasses of a similar composition were altered by aqueous alteration to yield the current phyllosilicate dominated compositions of petrographic type 1 and 2 carbonaceous chondrites[21–24].

In addition to phyllosilicates, other minerals are also thought to form during aqueous alteration, such as carbonates, magnetite and Fe-sulfides[21]. Carbonates in particular have been hypothesised to form through oxidation of organic matter[25]. However, this process cannot account for their entire occurrence in carbonaceous chondrite meteorites, with their generation from $CO_2$ found within aqueous fluids a likely source[26–28]. The $CO_2$ was likely accreted within ices that may have possessed a comet-like composition, consisting of largely water, but with significant amounts of ammonia, CO, $CO_2$ and simple organic compounds, such as formaldehyde, acetamide, methylisocyanate, acetone and formic acid[29,30].

The formation pathways for amino acids generated through aqueous alteration have not been characterised in their entirety, but Strecker synthesis, involving ammonia, ketones and aldehydes has been proposed to form some α-amino acids[31], while Michael addition of ammonia to α,β-unsaturated nitriles, with subsequent hydrolysis has been suggested as the pathway to form β-amino acids[32]. More recently, a number of hydrothermal experiments have been found to produce an array of α-, β- and γ-amino acids from fluids with a comet-like ice composition[29,33,34].

Despite isotopic and enantiomeric data indicating an extraterrestrial origin for the amino acids detected within carbonaceous chondrites, concerns still remain regarding the contribution of terrestrial contamination[4,35]. In order to minimise such contamination and shed new light on the origin and evolution of the building blocks of life contained within extraterrestrial objects, samples were returned from the Ryugu asteroid by the Hayabusa2 mission and made available for scientific analysis[36].

Here, two Ryugu particles from different touchdown sites (TD), A0022 (TD1) and C0008 (TD2), were analysed by ultrahigh performance liquid chromatography-Orbitrap-mass spectrometry (UHPLC-OT-MS) in order to detect and quantify their amino acid inventories. The two TD were situated on the equatorial ridge of Ryugu and ~870 m apart from one another[14,37,38]. The first touchdown is thought to have sampled material at the very surface of Ryugu, while the second touchdown is thought to have sampled material ejected from a nearby artificial impact crater (maximum depth ~1.7 m) and thus represent both surface and sub-surface material[14,36,38]. In particular, the second touchdown site showed a darker albedo relating to the artificial impact crater and the surrounding ejected material, when compared to the regions beyond the impact crater[38]. Such an observation could relate to the alteration of organic matter, which becomes brighter on irradiation, due to the formation of graphite at the expense of the original organic matter[20,39].

Previously, it was found that some of the TD1 particles, including A0022, were more affected by irradiation than others, including C0008 and many other TD2 particles[14]. In the aforementioned study, Raman spectroscopy demonstrated that for A0022 the G-band, arising from organic matter, had a lower peak centre position and a larger full width at half maximum (FWHM) value, compared to C0008. Such differences in the G band can be attributed the effects of irradiation, which act to lower the G-band peak centre position and increase the G-band FWHM[40]. Interestingly, desorption electrospray ionisation-orbitrap-mass spectrometry (DESI-OT-MS), which was applied to C0008 and a different TD1 particle to that mentioned above (A0048), showed that the intensity and distribution of soluble organic matter (SOM) in the TD2 particle were greater compared to in the TD1 particle[14]. Again, this was interpreted as being due to the destruction of the SOM in the TD1 particle by irradiation.

While the amino acids in C0008 were described previously[14], no quantitation was performed in that study. With the amino acid concentration data reported here for both a TD1 and TD2 sample, the relationships between amino acids from different Ryugu samples and other extraterrestrial samples will be elucidated and the formation pathways affecting these amino acids better constrained. Such information will then be used to better understand the processes operating on Ryugu and other C-type asteroids.

## Results and discussion
### Ryugu amino acid concentrationsn
The Ryugu amino acids were detected using a newly developed highly sensitive analytical technique[41] (Fig. 1) and quantitation was subsequently undertaken. Tandem mass spectrometry ($MS^2$) data was recorded when the amino acids yielded sufficient intensity (Supplementary Figs. 1–3). While the amino acid concentrations of the two particles were different, no uniform difference in the amino acid abundances between A0022 and C0008 were observed (Table 1 and Fig. 2). For instance, the amino acids were not found to be all higher in one sample than the other. Such an observation is interesting, because TD1 samples were found to have been more effected by irradiation than the TD2 samples, likely as a result of being closer to the surface of present day Ryugu and thus experienced greater solar wind and cosmic ray fluxes[14]. As such, it was expected that irradiation may have more readily destroyed the amino acids in A0022 compared to C0008 and thus the latter might contain uniformly higher amino acid abundances.

Instead, while glycine, aspartic acid and valine were at higher concentration in C0008, β-alanine, serine and glutamic acid were higher in A0022. Meanwhile, various amino acids could be quantitated in one particle, but not the other. Of particular interest is N,N-

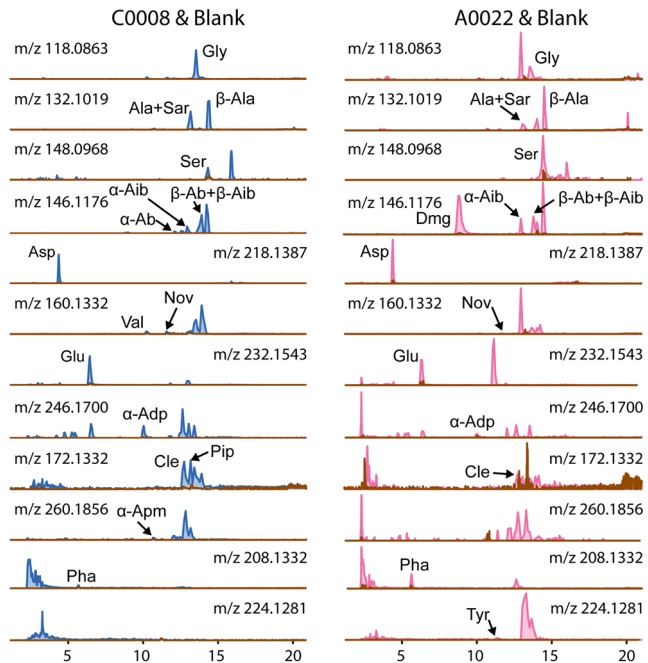

**Fig. 1 | Extracted ion chromatograms of the amino acids detected within C0008 (blue) and A0022 (pink).** The brown trace indicates the blank and it is clear that in all cases the response from the sample is higher than that of the blank. See Table 1 for the codes referring to the amino acids.

dimethylglycine, which was the most abundant amino acid in A0022, but being below detection limit in C0008. While the weights of each Ryugu particle aliquot analysed were different (C0008 = 1.900 mg and A0022 = 0.988 mg), as C0008 weighed more this cannot explain why N,N-dimethyl glycine is not detected in C0008.

Another interesting observation is the different concentrations of glycine and β-alanine observed in A0022 and C0008. Accordingly, the corresponding β-alanine/glycine ratio for the two particles is significantly different, with C0008 having a ratio of 0.57 and A0022 a ratio of 1.03. For comparison Orgueil (CI1) was indicated to have a range of between 1.71 and 3.16 and Murchison (CM2) of between 0.26 and 0.43[16,41,42]. The β-alanine/glycine ratio has been previously demonstrated to be sensitive to the degree of aqueous alteration experienced on a given meteorite's parent body, with higher levels of aqueous alteration yielding higher ratios[43,44]. The β-alanine/glycine ratio is thought to be affected either by a greater contribution of α-amino acids, including glycine, via Strecker synthesis during low levels of aqueous alteration or by a heightened destruction of α-amino acids via oxidation during high levels of aqueous alteration[43]. As such, although the Ryugu sample sizes utilised here are small and may thus not be entirely representative of the bulk asteroid, their β-alanine/glycine ratios may suggest that Ryugu experienced more intense aqueous alteration compared to Murchison (CM2), but less intense aqueous alteration compared with Orgueil (CI1). Based on the β-alanine/glycine ratio, there is also an apparent difference in the aqueous alteration experienced between the two particles, with A0022 having experienced higher levels of aqueous alteration. Such a finding indicates that planetesimals likely record variable levels of aqueous alteration, which is in agreement with another Ryugu study that found delicate presolar material survived in some small-scale regions of Ryugu due to distinctly lower levels of aqueous alteration[45].

Furthermore, while the current study utilises small sample sizes, which may not entirely capture the bulk β-alanine/glycine ratio of TD1 and TD2 on Ryugu, the disparity between the ratios of the two particles reported here indicates that at least on a local (sub-mm) scale aqueous alteration levels show heterogeneity. Indeed, it has been suggested

that aqueous alteration on meteorite parent bodies may have been static and occurred in many localised isolated closed systems[46,47], at least for bodies <40 km in radius, which would have prevented elemental fractionation. If the localised systems accreted different quantities or compositions of ice this may explain the differing levels of aqueous alteration among the Ryugu particles.

## The formation and evolution of Ryugu amino acids
As mentioned above, glycine, β-Alanine and serine are all major components in residues resulting from the UV processing of comet-like ices[7–9] (Fig. 3, processes 1a and 1b). While it may be possible to generate all the amino acids found in carbonaceous chondrites in such experiments and at the right abundances, no experiments have yet achieved this. Nevertheless, it is likely that some component of the aforementioned amino acids produced in the ISM/PSN would have been accreted into Ryugu, if it formed in the outer solar system as its Cr and Ca isotopes suggest[14,15]. While N,N-dimethylglycine may have been found previously in the Murchison meteorite (CM2)[48], it was not definitively detected and thus not quantified. Nevertheless, if sarcosine (N-methylglycine) is a common product in irradiated ice experiments[8], then it is likely that N,N-dimethylglycine can be produced as well. Therefore, it is possible that at least some of the amino acids were produced in the ISM or outer PSN before their accretion by Ryugu (Fig. 3, process 2).

In such a scenario, A0022 would be required to accrete larger quantities of ice (Fig. 3, process 3), compared to C0008 (Fig. 3, process 4), in order to provide the high concentration of N,N-dimethylglycine in A0022. The ice would have likely been composed of $H_2O$, $CO_2$ and other volatiles, such as ammonia and simple organic molecules[30]. Indeed, the higher abundances of carbonates (6.9 vol% for A0022 and 1.65 vol% for C0008), which could have formed from $CO_2$[26], the higher porosity (51 vol% for A0022 and 44 vol% for C0008) that may have housed ice and the higher concentrations of β-Alanine and serine in A0022 support the above theory. Moreover, the higher β-alanine/glycine ratio for A0022 is also consistent with the accretion of higher abundances of ice, compared to C0008, which would have led to higher levels of aqueous alteration.

However, glycine, a common product of comet-like ice irradiation experiments[7,8] is not higher in A0022 compared to C0008. One explanation, is that glycine was accreted in much higher concentrations and subsequently reacted to produce N,N-dimethylglycine (Fig. 3, process 5). Indeed, N,N-dimethylglycine is produced commercially through the reaction of aqueous formic acid and formaldehyde with glycine via a reductive amination termed the Eschweiler–Clarke reaction[49]. Accordingly, both formaldehyde and formic acid were observed in the comet 67 P/Churyumov–Gerasimenko[30] and could thus be expected to be in ice accreted by Ryugu. Furthermore, during the Eschweiler–Clarke reaction $CO_2$ is also produced, which could further contribute to the higher abundance of carbonate found in A0022, compared to C0008.

Alternatively, synthesis of many of the amino acids could have proceeded via Strecker synthesis and Michael addition, with subsequent reactions to yield specific amino acids. During such processes, the presence of water is essential and if formic acid and formaldehyde were present, then N,N-dimethylglycine could be produced at the expense of glycine (Fig. 3, process 5), a major product of Strecker synthesis[5]. If higher ice abundances were accreted by A0022, than C0008, then the higher quantity of N,N-dimethylglycine and other amino acids in the former could be explained. Additionally, as higher levels of aqueous alteration are likely to yield higher abundances of secondary alteration minerals, the higher abundances of carbonate, magnetite, coarse Fe-sulfides and trace elements in A0022 may also be explained[14]. As such, production of the amino acids solely on the Ryugu progenitor planetesimal may also be able to explain the concentrations reported here and the results of previous geochemical analyses.

**Table 1 | Blank corrected concentrations of amino acids in the Ryugu particle A0022 and C0008**

| C# | Code | Compound Name | C0008 (TW) Conc. ng/g | | A0022 (TW) Conc. ng/g | | Orgueil (P2023) Conc. ng/g | | Orgueil (G2010) Conc. ng/g | |
|---|---|---|---|---|---|---|---|---|---|---|
| | | | Average | SD | Average | SD | Average | SD | Average | SD |
| C2 | Gly | Glycine (α) | $3.0 \times 10^3$ | 32 | $2.3 \times 10^3$ | 18 | $1.5 \times 10^3$ | 115 | 865 | 450 |
| C3 | β-Ala | β-Alanine | $1.7 \times 10^3$ | 64 | $2.4 \times 10^3$ | 70 | $2.5 \times 10^3$ | 175 | $2.7 \times 10^3$ | 675 |
| C3 | Ser | Serine (α) | 257 | 8 | $1.0 \times 10^3$ | 99 | BDL | BDL | <1 | – |
| C4 | Thr | Threonine (α) | BQL | BQL | BQL | BQL | 643 | 11 | – | – |
| C4 | Dmg | N,N-Dimethylglycine (α) | BDL | BDL | $5.8 \times 10^3$ | 33 | BDL* | BDL* | – | – |
| C4 | α-Ab | α-Aminobutyric Acid | 132 | 18 | BDL | BDL | 146 | 9 | 71 | 49 |
| C4 | α-Aib | α-Aminoisobutyric Acid | 290 | 22 | BQL | BQL | 400 | 60 | 343 | 140 |
| C4 | Asp | Aspartic Acid (α) | 288 | 3 | 220 | 23 | 471 | 17 | 109 | 59 |
| C5 | Val | Valine (α) | 60 | 5 | 45 | 6 | BDL | BDL | – | – |
| C5 | Nov | Norvaline (α) | 78 | 12 | BQL | BQL | BDL | BDL | – | – |
| C5 | Iva | Isovaline (α) | 51 | 14 | BDL | BDL | BDL | BDL | – | – |
| C5 | Glu | Glutamic Acid (α) | 457 | 19 | 520 | 18 | 625 | 16 | 130 | 38 |
| C6 | α-Ada | α-Aminoadipic Acid | 70 | 4 | BQL | BQL | 117 | 17 | – | – |
| C6 | Cle | Cycloleucine (α) | 170 | 2 | BQL | BQL | 226 | 10 | – | – |
| C6 | Pip | Pipecolic Acid (α) | BQL | BQL | BDL | BDL | 743 | 146 | – | – |
| C7 | α-Apa | α-Aminopimelic Acid | BQL | BQL | BDL | BDL | 33 | 2 | – | – |
| C9 | Phe | Phenylalanine (α) | BDL | BDL | 85 | 7 | BDL | BDL | – | – |
| C9 | Tyr | Tyrosine (α) | BQL | BQL | 56 | 11 | BDL | BDL | – | – |

*Conc* concentration, *C#* carbon number, *TW* this work, P2023 = ref. [41], G2010 = ref. [42], *BDL* below detection limit, *BQL* below quantitation limit,—means no results were reported and * indicates the data was reinvestigated and the result was included here, but not published in the previous study. Amino acids with a (α) are alpha amino acids, while amino acids with an α or β at the start of their name are alpha or beta amino acids, respectively.

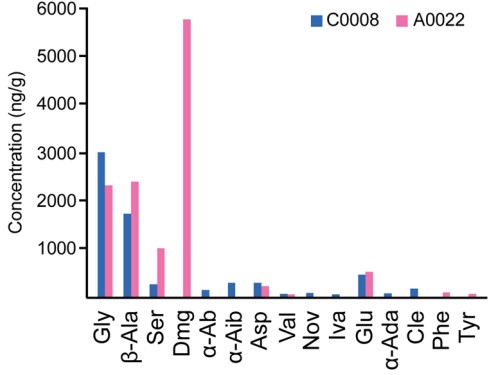

**Fig. 2 | A bar chart showing the concentration of amino acids in A0022 and C0008.** Note that the concentrations are different for the majority of amino acids between the two Ryugu particles. See Table 1 for the codes referring to the amino acid names.

In summary, if the Ryugu amino acids were predominantly synthesised on the Ryugu progenitor planetesimal, then it is likely that this would have occurred ~2.6 Ma after the formation of calcium–aluminium-rich inclusions, at a time when the accreted ice had melted to give liquid water rich fluids[14]. The differences in the amino acids thus representing differences in the abundances of ice (water/rock) accreted throughout Ryugu at a very small scale.

The findings reported here highlight the importance of a liquid aqueous media for the prebiotic synthesis of amino acids in extra-terrestrial samples and highlight the importance of sample returns in enabling the analysis of uncontaminated extraterrestrial samples. Furthermore, the results discussed here demonstrate that it is necessary to conduct a comprehensive analysis of both the organic and inorganic components of such samples, in order to investigate the source of the organic compound heterogeneity. For example, determining the

abundances of inorganic phases, such as carbonate, which forms from $CO_2$, and other secondary minerals associated with aqueous alteration (e.g. magnetite, sulfides and phyllosilicates) can highlight the different availabilities of volatiles and thus potentially yield information concerning the quantities of accreted ices. When comparing this with organic information, such as the β-alanine/glycine ratio or the abundances of potential alteration products (e.g. N,N-dimethylglycine) of primarily accreted amino acids (e.g. glycine), important information relating to the formation and evolution of organic matter on icey planetesimals can be obtained. In this way, heterogeneities in organic compounds, such as amino acids, can be used to better constrain the processes, which are ultimately responsible for the production of the building blocks of life. Such processes are responsible for shaping our solar system and likely making it habitable, and can thus provide insight into the potential habitability of other solar systems, as well as help to explain how life could have arisen on Earth.

## Methods

### Sample preparation and extraction

An aliquot of C0008 (1.900 mg) and A0022 (0.988 mg) were placed into sealable Teflon vials and weighed, along with two vials representing procedural blanks. The higher value for a given amino acid was used for blank correction. All tools and the vials were cleaned multiple times with once distilled ion exchanged (1DIE) water (Puric-ω, Organo Co.) and UGR grade (for trace analysis) MeOH (Kanto Chemical Co., Inc.) before use. Pipettes were heated to 500 °C to remove any organic contamination, but 10 ml vials were cleaned with solvents, because the heating step resulted in a higher probability of the vials breaking on freezing. The handling of the samples, extraction and analysis were all carried out in a class 10 clean room at the Pheasant Memorial Laboratory (PML) for geochemistry and cosmochemistry, Institute for Planetary Materials (IPM), Okayama university at Misasa. Note that here Teflon vials were used for extraction of the samples, because this allowed for better sealing than glass vials and demineralisation of the

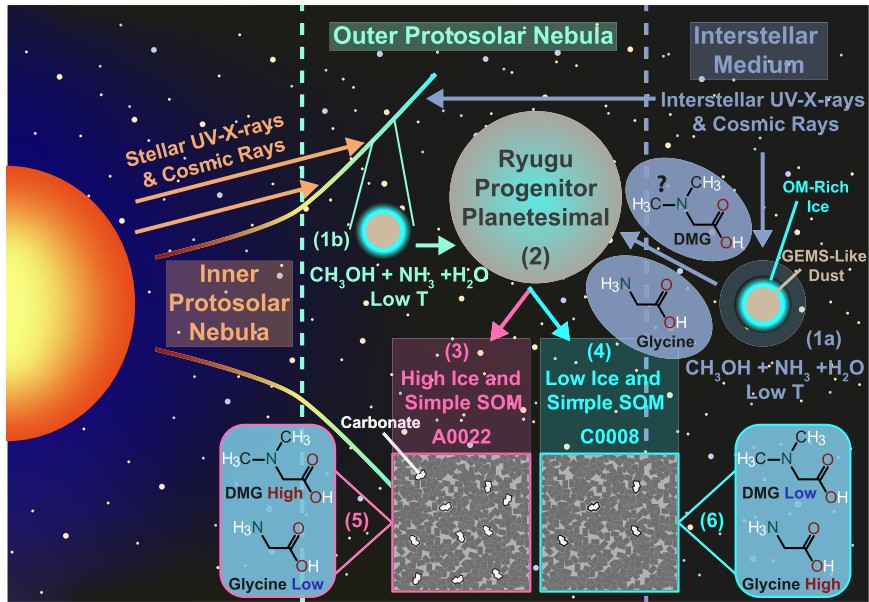

**Fig. 3 | An illustration of the different processes that could explain the different concentrations of amino acids within the Ryugu particles A0022 and C0008.** 1a: formation of amino acids in the interstellar medium (ISM), 1b: formation of amino acids in the protosolar nebula (PSN), 2: accretion of ices and simple organic matter (including amino acids) into the Ryugu progenitor planetesimal, 3: accretion of higher amounts of ice and their organic cargo in A0022, 4: accretion of lower amounts of ice and their organic cargo in C0008, 5: formation of higher levels of N,N-dimethylglycine (DMG) in A0022 at the expense of glycine and 6: lower levels of DMG are formed due to a lower level of accreted ice components meaning less glycine was able to react with other ISM/PSN ice components during lower levels of aqueous alteration, compared to A0022. T temperature, OM organic matter, SOM soluble organic matter, GEMS glass with embedded metal and sulfides and DMG N,N-dimethylglycine.

residues by HF for isolation of the IOM and further analysis, which is reported elsewhere[14].

In light of the small amount of Ryugu sample returned to Earth (5.4 g) and the even smaller amount of material available for any given study, the authors of this study developed a new technique to detect amino acids in extremely small sample sizes (~1–2 mg)[41]. The extraction of the samples involved heating in 300 μL of 1DIE water within the Teflon vials for 20 h at 110 °C. Note that the Ryugu samples were very fragile and completely broke up on addition to water, so no grinding or sonication was necessary. After cooling, the supernatant was removed, along with 3x 1DIE water washes of the Ryugu residues, and transferred to glass 10 ml vials, which were sealed and frozen overnight at −60 °C. Subsequently, the frozen extracts were freeze dried and derivatised with 300 μL 2 M HCl/isopropyl alcohol at 110 °C, to yield AA isopropyl esters (AAIE). The derivatised extracts were then reduced under $N_2$, dissolved in water and freeze dried again to ensure the complete removal of water. Ethyl acetate (100 μL) was added to dissolve the AAIE and the extracts transferred to 300 μL glass insert vials.

### UHPLC-OT-MS analysis

The amino acids were analysed using a ThermoFisher Scientific Accucore™ 150 Amide HILIC column in reverse phase mode on a Vanquish™ UHPLC unit. The UHPLC is a binary system and separation was conducted using two solvents: phase A was 10 mM ammonium formate in 1DIE water (adjusted to pH 3.5 using formic acid) and phase B was 100% ACN. The gradient was adjusted from 100% B to 79% B over 15 min, to 0% B over a further 5 min, then held at 0% B for 5 min, before increasing to 100% B over 5 min and holding at 100% B for an additional 5 min. A 0.15 mL/min flow rate and a column compartment temperature of 30 °C were used for UHPLC separation.

The UHPLC unit was coupled to an Orbitrap Fusion Mass Spectrometer (Thermo Scientific). The following parameters were used for the UHPLC-OT-MS analyses: an ion transfer tube and vaporiser temperature of 300 °C, positive ion voltage of 3500 V, sheath gas of 50,

auxiliary gas of 15, RF lens of 55%, m/z range of 50–500, OT resolution of 240,000 and an AGC target of $2 \times 10^5$.

Amino acids were identified and quantitated using an in-house standard containing the amino acids of interest at concentrations ranging from 1 to 0.00075 μg g$^{-1}$. For more information concerning the validation and quantitation of the amino acids detected here, please see the supplementary information and the previously published methods paper[41].

## Data availability

Amino acid concentration source data are included within Table 1 of this paper and the data used to generate the extracted ion chromatogram traces in Fig. 1 have been deposited in the Open Science Framework repository https://doi.org/10.17605/OSF.IO/HBEJS[50].

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

## Acknowledgements

We are greatly indebted to PML members for their assistance in maintaining the laboratory, especially Kayo Tanaka and Yukari Shimizu whose daily efforts greatly aided the work carried out here. This work was supported by Ministry of Education, Culture and Sports, Science and Technology (MEXT) of Japan.

## Author contributions

C.P. conceived the study, collected and interpreted the data and wrote the manuscript. T.Ot., M.Ya. and C.S. assisted with sample preparation, extraction and the collecting of the data. K.K. aided with data processing and interpretation. R.T., T.K., H.K. and E.N. helped with interpretation and improving the manuscript. E.N. also helped with the conception of the study and procuring funding for the work. M.A., A.M., A.N., S.N., M.N., T.Ok., T.S., S.T., F.T., Y.T., T.U., S.W., T.Y., K.Y. and M.Yo. were involved with the Hayabusa2 mission which collected the samples analysed here from the asteroid Ryugu and undertook preliminary characterisation of the samples once they had arrived on Earth. All authors were involved in the preparation of the manuscript.

## Competing interests

The authors declare no competing interests.
