## [Peer Review File · Nature Communications]

Insights into the formation and evolution of extraterrestrial amino acids from the asteroid RyuguReviewer #1 (Remarks to the Author):

The present manuscript reports on the UHPLC analysis of amino acids in A0022 and C0008 grains from the Ryugu asteroid. Based on different composition/concentrations of amino acids in 0.998 mg and 1.900 mg fragments, the authors attempt to infer their formation history. The authors analyse very small quantities of the Ryugu asteroid (~1 and ~2 mg), which is indeed methodologically very demanding and interesting, especially considering the analysis of such precious samples. The methodology is not reported here for the first time but was already reported in an earlier publication, which is currently under review in another journal and is enclosed as a supplementary document to the present manuscript.

The authors repeatedly highlight in the manuscript that the analysis of return samples is extremely important since terrestrial contamination remains an issue in the analyses of meteorites. However, the authors at the same time report on contaminations in their procedural blanks, which diminishes the strength of the interpretations drawn from the analysis of the apparently purest extra-terrestrial samples. Obviously, the limited amount of sample did not allow to do an isotopic ratio analysis and hence real origin of the protein amino acids identified cannot be confirmed. The enantiomers of chiral amino acids were also not resolved and hence the extra-terrestrial origin of protein amino acids could not be confirmed on the basis of the presence of the non-biological D-enantiomers either. Was it not possible to avoid the contamination? Was any investigation carried out to reveal the source of contamination and if so any effort made to eliminate it? The authors mention that several procedural blanks were prepared, it is, however, not clearly specified how many, whether the level of contamination was consistent between them and whether they subtracted the average or maximum procedural blanks signal from the samples signal. In any case, drawing any conclusions from protein amino acids contents in the limited amount of sample they had seems problematic since the contamination issue cannot be fully disregarded. Why the glassware used for the analysis of samples was not burnt at high temperatures to eliminate biological contamination? This is a common practise in meteoritic sample analyses. Were any steps taken to facilitate liberation of amino acids from the grains, e.g. crushing or sonication?

Can ~1 and ~2 mg samples be representative for drawing conclusions about the formation history of the organics in Ryugu? The authors do not cite the first literature on Ryugu analysis by Parker et al. (LPSC, 2022), where amino acids and their concentration in 13.08 mg of A0106 Ryugu sample (much bigger piece of the asteroid) were reported. Even though the A0106 (Parker et al.) and A0022 (present manuscript) samples come from the same Chamber A, i.e. the samples collected during the first touch-down sampling, their composition is different. Notably, in the present manuscript authors extensively discuss the ratio of β -alanine/glycine and draw implications about the extent of aqueous alteration that Ryugu would have experienced. For C0008 they report this ratio to be 0.57 and for A0022 1.04. Since these are between 0.26–0.43 in Murchison and 2.57–3.16 in Orgueil, they conclude that Ryugu must have experienced more intense aqueous alteration than Murchison and less intense than Orgueil. However, the A0106 sample analysed by Parker et al. has a β -alanine/glycine ratio of 7.14, i.e. significantly different even though this sample should be from the same collection site as A0022 from the present study. The authors argue that the β -alanine/glycine ratio is significantly different between A0022 and C0008, however, there is apparently greater difference between A0022 and A0106. The values of 2.54 and 3.12 for Orgueil in the present study are not far from the literature range (2.57–3.16), thence the sample treatment and derivatisation procedure should not have played a great role in affecting this ratio? Interestingly, N,N-dimethylglycine, the most abundant amino acid in the A0022 sample was not reported by Parker et al., possibly due to different sample extraction, derivatisation and analytical protocol.

In addition to the hot water hydrolysis, further HCl hydrolysis of the samples could have proceeded during the derivatisation protocol. Did the authors consider potential ester hydrolysis of the derivatives during the HPLC analysis given an acidic mobile phase was employed?

Authors argue in the Introduction section that so far only extra-terrestrial sources have been found to contain amino acids with an L-excess. This is, however, not a fair claim, since another source of amino acids could have been on the early Earth, however, the detection of an L-excess of these is obviously virtually impossible in the present-day biosphere, not meaning there was none.

Furthermore, the authors argue that only simple amino acids can be produced in simulated interstellar ice experiments and more complex amino acids require aqueous processing. It is, however, ambiguous what the authors mean by "complex". At least 26 different amino/diamino acids were identified in simulated interstellar ices (water, methanol, ammonia) in the study by Meinert et al., *ChemPlusChem*, 2012. Moreover, the molecular inventory produced in the interstellar ices very much depends on the starting material and hence it does not mean that if certain amino acids were yet not detected in the simulated ices, they cannot be formed there.

Authors argue that many amino acids could be detected in one sample, but not the other (including protein amino acids which could be at least partly due to contamination). However, one has to keep in mind that the A0022 was only about half mass of the C0008 sample, with such small quantities and expectable sample inhomogeneity this is in my view not enough to draw any concrete conclusions about differences in formation history based on the differences in composition of these "single-shot" experiments from two different collection sites.

In summary, the analysis of such small sample quantities seems very useful for a first-shot analysis to get an idea about what compounds are expected to be present in a new yet-uncharacterised sample, however, it does not seem the method of choice for concentration determination due to limited signal intensities in combination with potential contamination, which cannot be fully excluded. In addition, such small fragments cannot be a good representation of the inhomogeneous asteroid body and hence conclusions about the formation history of the amino acids in Ryugu based on analysing such small fragments are very premature. Finally, even when taking the two particles' amino acid composition as they are, the authors can only hypothesize about their origin, especially the authors in their interpretations rely on an ambiguous statement that only a limited selection of simple amino acids can be formed in simulated interstellar ices.

Reviewer #2 (Remarks to the Author):

I have eagerly awaited data and analyses that help to contextualize the chemistry observed on the asteroid Ryugu so I naturally feel these are noteworthy results. While the dataset presented is modest in size the text did a very good job of presenting the different mechanisms that can be invoked by observed patterns. I think the value of this manuscript will only increase as the field pieces together the voluminous array of data available from the mapping of the asteroid. While I found the methodology to be sound it is unfortunate that the sample preparation method is still in review.

While the manuscript reads well I do think it would be good to include a general description of aqueous alteration since it is an important pathway for this compound assemblage. It would also be good to for the authors to describe a "comet-like ice composition" (Line 51) since it includes many solids besides water ice.

Line 57: Is there an estimate of the depth of particle C0008 or an estimate of its proximity to A0022? Most of the paper focuses on the chemical differences between these samples but with little description of the known heterogeneity of the body from other datasets or an indication of how close these two sites are it is difficult to understand how reaction space might change. This also relates to comment below.

Line 74: I know space is often tight on these papers but it would be useful to have more explicit about the flux estimates from the Nakamura paper. Relative irradiation amounts would be interesting to consider given that the proximity of the samples and the sampling depth at TD2 are not described.

Line 81: In lieu of structures it would be helpful to include an alpha, beta designation for each species to indicate potential reaction pathways.

Line 97: The authors have accidentally labelled both ratios as A0022.

Line 113: It would be nice to have a description of the formation of Ryugu earlier in the paper. The manuscript cites Nakamura repeatedly but just a sentence or two to describe the isotope data for the formation location would help (as well as a mention about irradiation).

Line 114: This sentence is confusing. Are the authors saying that N,N-dimethylglycine has been detected but not quantified? Could the authors at least say what petrographic class these specimens are, given the importance of aqueous alteration?

Line 128: I found this figure very helpful but it would be more illustrative if each mechanisms was numbered and then that number indicated in the accompanying text.

Line 135: Appreciated the inclusion of comet data. Are the reaction rates for glycine Strecker formation, sarcosine formation in ices, and N,N-dimethylglycine (with formic acid and formaldehyde) known? Could these relative reaction rates (if known) help answer this question of dynamics and compound accumulation?

Line 142: Again referencing Nakamura but I'm not sure how higher ice abundance relates to higher abundance of carbonate, magnetite, coarse Fe-sulfides or trace elements. Please elaborate even if briefly.

Line 148: CAI is not defined.

Line 155: I would like to see this statement about the value of combined organic and inorganic analysis strengthened with a summary of the examples given in the paper.

In summary, I think just a few changes to bring context, clarify connections, and streamline the many interesting chemical pathways would make this a much easier read. The authors have maximized just a few chemical relationships to connect compounds to small body formation processes and the paper just needs to make sure these potential explanations make sense given the pre-existing datasets.

REVIEWER COMMENTS

Reviewer #1 (Remarks to the Author):

The present manuscript reports on the UHPLC analysis of amino acids in A0022 and C0008 grains from the Ryugu asteroid. Based on different composition/concentrations of amino acids in 0.998 mg and 1.900 mg fragments, the authors attempt to infer their formation history. The authors analyse very small quantities of the Ryugu asteroid (~1 and ~2 mg), which is indeed methodologically very demanding and interesting, especially considering the analysis of such precious samples. The methodology is not reported here for the first time but was already reported in an earlier publication, which is currently under review in another journal and is enclosed as a supplementary document to the present manuscript.

The authors repeatedly highlight in the manuscript that the analysis of return samples is extremely important since terrestrial contamination remains an issue in the analyses of meteorites. However, the authors at the same time report on contaminations in their procedural blanks, which diminishes the strength of the interpretations drawn from the analysis of the apparently purest extra-terrestrial samples. Obviously, the limited amount of sample did not allow to do an isotopic ratio analysis and hence real origin of the protein amino acids identified cannot be confirmed. The enantiomers of chiral amino acids were also not resolved and hence the extra-terrestrial origin of protein amino acids could not be confirmed on the basis of the presence of the non-biological D-enantiomers either. Was it not possible to avoid the contamination? Was any investigation carried out to reveal the source of contamination and if so any effort made to eliminate it?

Sorry, it is not totally clear whether the reviewer is referring to the Nature Communications (NC) article or the methods article (MA) enclosed for review purposes. Actually, in the NC article not much discussion has been given concerning the procedural blank contamination level, because the level is very low (as indicated by figure 1) and the values for the amino acids were blank corrected. However, you can in some cases see a peak in the blank for some amino acids. Many test runs were carried out in the run up to the analysis to determine the sources of contamination. No specific source was determined to be contributing all of this contamination, but it is highly likely that a very small background level of amino acids will be present in the lab. As our analyses are very sensitive, there will thus inevitably be a small amount of contamination present in many cases, but if the amino acids in the sample are high enough, which they are, then this should not be a problem. In all the cases where quantification was possible, you can see that the signal from the sample is much higher than that in the blank.

The authors mention that several procedural blanks were prepared, it is, however, not clearly specified how many, whether the level of contamination was consistent between them and whether they subtracted the average or maximum procedural blanks signal from the samples signal. In any case, drawing any conclusions from protein amino acids contents in the limited amount of sample they had seems problematic since the contamination issue cannot be fully disregarded.

Overall, 2 procedural blanks were run. The highest blank level for a given amino acid was used for the blank correction. Of course, it would have been great to have enough sample to examine the chirality of the amino acids, but unfortunately there are no available techniques that can do this for such small sample sizes. The samples have been curated in very clean environments, so really the only contamination should be from the lab environment and we used blanks to correct for this. As such, we concluded that the quantified amino acid values reported here should be realistic and thus conclusions can be drawn. We have included several sentences to clarify the blanks in the method text (Line 226-227).

Why the glassware used for the analysis of samples was not burnt at high temperatures to eliminate biological contamination? This is a common practise in meteoritic sample analyses.

Firstly, Teflon vials, after washing many washing steps, were used for the extraction protocol as they gave lower blank levels and were better at sealing in water vapour than small glass vials. Furthermore, demineralisation of the residue with HCl/HF to isolate IOM could then be proceeded in the same vial without the need to transfer the residue. As such, for the extraction vials no burning could be undertaken. For the 10 ml vials, it was noted that heating at elevated temperatures deformed the glass and made them worse at sealing in volatiles during derivatisation. Also, it was noted that the glass was weakened by this process and the tubes would sometime crack during freezing if they had been heated beforehand. We have now included several sentences to clarify this in text (lines 229-230 and 233-235).

Were any steps taken to facilitate liberation of amino acids from the grains, e.g. crushing or sonication?

We noticed that for Ryugu particles when they were put into the water, they effectively broke up into powder themselves without the need for crushing as they were so fragile. Orgueil does the same, but Murchison for example, would need to be either crushed or powdered using a microtome. We have now included some text in the methods section to explain this (lines 239-240).

Can ~1 and ~2 mg samples be representative for drawing conclusions about the formation history of the organics in Ryugu?

We don't claim that the samples are representative of the entirety of Ryugu. What we say is that they are representative of the particles from which they came and as we have the modal mineralogy for these particles, we can discuss the organic matter's formation history in the context of the mineralogy. I don't think any study on Ryugu can claim to be representative of the entire asteroid, as only 5.4 g was returned.

The authors do not cite the first literature on Ryugu analysis by Parker et al. (LPSC, 2022), where amino acids and their concentration in 13.08 mg of A0106 Ryugu sample (much bigger piece of the asteroid) were reported.

We did not cite Parker et al., 2022 as this is a conference abstract and presumably thus not peer reviewed. We understand that this will likely be published as an article at some point and at that point we would be happy to reference it in a future article.

We would also like to add that our group also reported our earlier findings in the Hayabusa2 symposium (2021), as did other groups, and we have not included any of those conference abstracts for the same reason.

Even though the A0106 (Parker et al.) and A0022 (present manuscript) samples come from the same Chamber A, i.e. the samples collected during the first touchdown sampling, their composition is different. Notably, in the present manuscript authors extensively discuss the ratio of β -alanine/glycine and draw implications about the extent of aqueous alteration that Ryugu would have experienced. For C0008 they report this ratio to be 0.57 and for A0022 1.04. Since these are between 0.26–0.43 in Murchison and 2.57–3.16 in Orgueil, they conclude that Ryugu must have experienced more intense aqueous alteration than Murchison and less intense than Orgueil. However, the A0106 sample analysed by Parker et al. has a β -alanine/glycine ratio of 7.14, i.e. significantly different even though this sample should be from the same collection site as A0022 from the present study. The authors argue that the β -alanine/glycine ratio is significantly different between A0022 and C0008, however, there is apparently greater difference between A0022 and A0106. The values of 2.54 and 3.12 for Orgueil in the present study are not far from the literature range (2.57–3.16), thence the sample treatment and derivatisation procedure should not have played a great role in affecting this ratio? Interestingly, N,N-dimethylglycine, the most abundant amino acid in the A0022 sample was not reported by Parker et al., possibly due to different sample extraction, derivatisation and analytical protocol.

Since the data of A0106 has not been published in a peer reviewed paper, we cannot evaluate why A0106 has a distinct β -alanine/glycine ratio at this moment. It is possible that some of the Chamber A particles, being on the surface of Ryugu, could be from exogenous micrometeorite impacts. Luckily, for our A0022 particle we have modal mineralogy data as we conducted a comprehensive geochemical analysis of our particles. As such we are able to confirm that it has a C11-like composition and is thus most likely from Ryugu as it is similar in that respect to all of our 16 Ryugu particles from both touchdown sites that we analysed.

We agree that the procedure should not have played a major role in affecting the data. We agree that the HPLC procedure and the type of derivatisation we employed will mean that the amino acids we can detect and their sensitivities will likely be different to the Parker study and that may explain this.

In addition to the hot water hydrolysis, further HCl hydrolysis of the samples could have proceeded during the derivatisation protocol. Did the authors consider potential ester hydrolysis of the derivatives during the HPLC analysis given an acidic mobile phase was employed?

We are not sure we completely understand. Are you saying that the amino acids derivatised as isopropyl esters could break down somewhat during the HPLC method as water reacts with the isopropyl esters? Or that the formic acid could form further esters from unreacted amino acids during the HPLC method? If so, then this shouldn't affect our results, because our standard was derivatised in the same way and run through the same method, so it would have been affected in the same way as the sample.

Authors argue in the Introduction section that so far only extra-terrestrial sources have been found to contain amino acids with an L-excess. This is, however, not a fair claim, since another source of amino acids could have been on the early Earth, however, the detection of an L-excess of these is obviously virtually impossible in the present-day biosphere, not meaning there was none.

We don't say that there were no potential environments on the Early Earth that could have generated L-excesses of amino acids, just that the only environments known to contain these are extraterrestrial ones. No one has so far proposed a plausible way to produce L-excesses of amino acids on the early Earth. As such, we don't think what we have said is wrong or misleading.

Furthermore, the authors argue that only simple amino acids can be produced in simulated interstellar ice experiments and more complex amino acids require aqueous processing. It is, however, ambiguous what the authors mean by "complex". At least 26 different amino/diamino acids were identified in simulated interstellar ices (water, methanol, ammonia) in the study by Meinert et al., ChemPlusChem, 2012. Moreover, the molecular inventory produced in the interstellar ices very much depends on the starting material and hence it does not mean that if certain amino acids were yet not detected in the simulated ices, they cannot be formed there.

Meinert et al., 2012 does not quantify their amino acids, so it is hard to understand if like other studies (e.g. Modica et al., 2018 and Oba et al., 2016) many of the longer chain length or larger mass amino acids (more complex) are only present as small amounts compared to say glycine, alanine and β -alanine. No interstellar ice analogue study has so far demonstrated that they can produce amino acids at carbonaceous chondrite abundances. As such, aqueous alteration on meteorite parent bodies has been suggested (e.g. Modica et al., 2018, Martins et al., 2015, Glavin et al., 2010) to be responsible for increasing all the other amino acids relative to Glycine. As far as we know glycine is usually the most abundant amino acid in interstellar ice analogue studies and even β -alanine is low compared to it, being quite a bit lower than in most CM, let alone CI chondrites. The Paris meteorite is the closest in terms of the β -alanine/glycine among the CMs (e.g. Modica et al., 2018). As such, in terms of the experiments so far undertaken, we don't think it is misleading to say that aqueous alteration is likely required to explain the abundances of the amino acids in Ryugu, being that it is of a CI-like composition (highly aqueously altered). We have now clarified meaning of simple and complex amino acids in text (Line: 56-58).

Authors argue that many amino acids could be detected in one sample, but not the other (including protein amino acids which could be at least partly due to contamination). However, one has to keep in mind that the A0022 was only about half mass of the C0008 sample, with such small quantities and expectable sample inhomogeneity this is in my view not enough to draw any concrete conclusions about differences in formation history based on the differences in composition of these "single-shot" experiments from two different collection sites.

We do not use any of the low abundance amino acids to draw any formation history conclusions. We use the presence of DMG in A0022 and the β -alanine/glycine

mostly. These amino acids are present at quite high abundances. We make mention of the differences in amino acid abundance between the two samples, but for the exact reason highlighted above by the reviewer we do not focus on the differences of the lower abundance amino acids. This again should alleviate any contamination concerns as well, because the main implications of the paper rest on some of the most abundant amino acids, which had some of the least contamination when looking at the blank traces in figure 1.

In summary, the analysis of such small sample quantities seems very useful for a first-shot analysis to get an idea about what compounds are expected to be present in a new yet-uncharacterised sample, however, it does not seem the method of choice for concentration determination due to limited signal intensities in combination with potential contamination, which cannot be fully excluded. In addition, such small fragments cannot be a good representation of the inhomogeneous asteroid body and hence conclusions about the formation history of the amino acids in Ryugu based on analysing such small fragments are very premature. Finally, even when taking the two particles' amino acid composition as they are, the authors can only hypothesize about their origin, especially the authors in their interpretations rely on an ambiguous statement that only a limited selection of simple amino acids can be formed in simulated interstellar ices.

We have to disagree with the reviewer and we feel that the reasons we have mentioned above more than justify why we believe our methodology is appropriate and the resultant findings provide a novel and interesting contribution to the scientific literature.

Reviewer #2 (Remarks to the Author):

I have eagerly awaited data and analyses that help to contextualize the chemistry observed on the asteroid Ryugu so I naturally feel these are noteworthy results. While the dataset presented is modest in size the text did a very good job of presenting the different mechanisms that can be invoked by observed patterns. I think the value of this manuscript will only increase as the field pieces together the voluminous array of data available from the mapping of the asteroid. While I found the methodology to be sound it is unfortunate that the sample preparation method is still in review.

We agree with the reviewer that it would be nice to have the methods paper accepted. Unfortunately, ACS Earth and Space Chemistry had a lot of problem with securing reviewers and the process has taken a long time, but we expect that the article should be accepted shortly.

While the manuscript reads well I do think it would be good to include a general description of aqueous alteration since it is an important pathway for this compound assemblage. It would also be good to for the authors to describe a "comet-like ice composition" (Line 51) since it includes many solids besides water ice.

Several paragraphs have now been added to the introduction section to describe aqueous alteration and a comet like ice composition (Lines: 59-74)

Line 57: Is there an estimate of the depth of particle C0008 or an estimate of its proximity to A0022? Most of the paper focuses on the chemical differences between these samples but with little description of the known heterogeneity of the body from other datasets or an indication of how close these two sites are it is difficult to understand how reaction space might change. This also relates to comment below.

The depth is hard to calculate, because the Ne isotope composition of the Ryugu particles likely records some input from ancient sources, when the dust grains were exposed to various irradiation sources prior to their accretion into Ryugu. As such, we can't make good cosmic ray exposure age estimates, which could be used to estimate the depth of the particles within the body. However, the crater from the second touchdown site was around 1.7 m in depth, so the maximum depth that C0008 could have come from is around 1.7 m. The touchdown sites were approximately 870 m apart. Accordingly, it is quite possible that there could be heterogeneity of the mineralogy and accreted materials, when they are from distinct regions on Ryugu. Also, if C0008 was from the deeper part of the crater, then this would be more than enough to generate a difference in the irradiation experienced from cosmic rays and thus account for the differences observed between A and C particles for some of the organic matter in previous studies. Some extra details of where the samples are from has been added to the introduction (Lines: 88-94).

Line 74: I know space is often tight on these papers but it would be useful to have more explicit about the flux estimates from the Nakamura paper. Relative irradiation amounts would be interesting to consider given that the proximity of the samples and the sampling depth at TD2 are not described.

So as mentioned above, it is difficult to separate the inherited Ne isotopes from irradiation exposure before the accretion of Ryugu and that generated by solar cosmic rays on the surface of Ryugu. As such it is hard to estimate the irradiation fluxes. When we estimated it, both A0022 and C0008 have similar fluxes, but it is not clear what their original inherited contributions would have been. When we use the Raman G band parameters for the IOM, we see that A0022 has a lower G-band peak centre and larger FWHM value, compared to C0008, which may indicate it has experienced more irradiation as this kind of difference was observed between organic soots that had be irradiated and those that had not. More description of this has been added to the introduction (Lines: 92-105).

Line 81: In lieu of structures it would be helpful to include an alpha, beta designation for each species to indicate potential reaction pathways.

We have now included this in table 1.

Line 97: The authors have accidentally labelled both ratios as A0022.

Thank you for pointing this out, we have now rectified this.

Line 113: It would be nice to have a description of the formation of Ryugu earlier in the paper. The manuscript cites Nakamura repeatedly but just a sentence or two to

describe the isotope data for the formation location would help (as well as a mention about irradiation).

We have now put an entire paragraph in the introduction about the formation of Ryugu (Lines: 45-55).

Line 114: This sentence is confusing. Are the authors saying that N,N-dimethylglycine has been detected but not quantified? Could the authors at least say what petrographic class these specimens are, given the importance of aqueous alteration?

The problem is that several review papers by Glavin and co-authors have included this amino acid as having been detected in meteorites, but the references are incorrect and none of them actually lead to a paper that even mentions the amino acid. The only reference of the amino acid that I could find in the literature, was from Lawless 1973 in GCA, where the author says they may have detected it, but are unsure. We have now removed the Glavin et al. reference and changed this sentence to indicate that DMG may have been detected in the Murchison meteorite by a previous study, but the identification was not definitive and put the Lawless reference (Lines: 161-162).

Line 128: I found this figure very helpful but it would be more illustrative if each mechanisms was numbered and then that number indicated in the accompanying text.

We have now included numbers to the processes in the diagram and put these in text.

Line 135: Appreciated the inclusion of comet data. Are the reaction rates for glycine Strecker formation, sarcosine formation in ices, and N,N-dimethylglycine (with formic acid and formaldehyde) known? Could these relative reaction rates (if known) help answer this question of dynamics and compound accumulation?

There has been some work on the thermodynamics of glycine formation from Strecker synthesis, but this is going to massively depend on the presence or absence of mineral catalysts and the exact temperature conditions and fluid chemistry of the site of reaction in the asteroid. As such, I think this will be a very difficult problem to solve even just for glycine. In terms of the other two, it may be possible to calculate a reaction rate for sarcosine, but I cannot find any literature values for it. As for DMG, I think again the exact conditions and catalysts involved will really affect the reaction rate. I think experiments would need to be carried out to determine these with plausible asteroidal parent body conditions, which is not within the scope of the current work, but would make for exciting future work.

Line 142: Again referencing Nakamura but I'm not sure how higher ice abundance relates to higher abundance of carbonate, magnetite, coarse Fe-sulfides or trace elements. Please elaborate even if briefly.

Several paragraphs have been added to the introduction to help explain aqueous alteration, including reference to the mineral phases highlighted by the reviewer and

also the text around the area referenced to by the reviewer has been updated to offer more clarity (Lines: 68-69, 198-199).

Line 148: CAI is not defined.

This has been rectified.

Line 155: I would like to see this statement about the value of combined organic and inorganic analysis strengthened with a summary of the examples given in the paper.

We have added some extra text to the last paragraph to help summarise the paper better (Lines: 212-219).

In summary, I think just a few changes to bring context, clarify connections, and streamline the many interesting chemical pathways would make this a much easier read. The authors have maximized just a few chemical relationships to connect compounds to small body formation processes and the paper just needs to make sure these potential explanations make sense given the pre-existing datasets.

We hope that our revisions have adequately addressed the reviewer's concerns. Please let us know if any further revisions are necessary.

Reviewer #1 (Remarks to the Author):

The comments are in the attached PDF.

Reviewer #1 Attachment on the following page

Answers to the authors' comments are below in green. Some of the comments are grouped together based on the similarity of their topic and a single answer is provided at the end of the grouped comments.

1)

The authors repeatedly highlight in the manuscript that the analysis of return samples is extremely important since terrestrial contamination remains an issue in the analyses of meteorites. However, the authors at the same time report on contaminations in their procedural blanks, which diminishes the strength of the interpretations drawn from the analysis of the apparently purest extra-terrestrial samples. Obviously, the limited amount of sample did not allow to do an isotopic ratio analysis and hence real origin of the protein amino acids identified cannot be confirmed. The enantiomers of chiral amino acids were also not resolved and hence the extra-terrestrial origin of protein amino acids could not be confirmed on the basis of the presence of the non-biological D-enantiomers either. Was it not possible to avoid the contamination? Was any investigation carried out to reveal the source of contamination and if so any effort made to eliminate it?

Sorry, it is not totally clear whether the reviewer is referring to the Nature Communications (NC) article or the methods article (MA) enclosed for review purposes. Actually, in the NC article not much discussion has been given concerning the procedural blank contamination level, because the level is very low (as indicated by figure 1) and the values for the amino acids were blank corrected. However, you can in some cases see a peak in the blank for some amino acids. Many test runs were carried out in the run up to the analysis to determine the sources of contamination. No specific source was determined to be contributing all of this contamination, but it is highly likely that a very small background level of amino acids will be present in the lab. As our analyses are very sensitive, there will thus inevitably be a small amount of contamination present in many cases, but if the amino acids in the sample are high enough, which they are, then this should not be a problem. In all the cases where quantification was possible, you can see that the signal from the sample is much higher than that in the blank.

The authors mention that several procedural blanks were prepared, it is, however, not clearly specified how many, whether the level of contamination was consistent between them and whether they subtracted the average or maximum procedural blanks signal from the samples signal. In any case, drawing any conclusions from protein amino acids contents in the limited amount of sample they had seems problematic since the contamination issue cannot be fully disregarded.

Overall, 2 procedural blanks were run. The highest blank level for a given amino acid was used for the blank correction. Of course, it would have been great to have enough sample to examine the chirality of the amino acids, but unfortunately there are no available techniques that can do this for such small sample sizes. The samples have been curated in very clean environments, so really the only contamination should be from the lab environment and we used blanks to correct for this. As such, we concluded that the quantified amino acid values reported here should be realistic and thus conclusions can be drawn. We have included several sentences to clarify the blanks in the method text (Line 226-227).

Why the glassware used for the analysis of samples was not burnt at high temperatures to eliminate biological contamination? This is a common practise in meteoritic sample analyses.

Firstly, Teflon vials, after washing many washing steps, were used for the extraction protocol as they gave lower blank levels and were better at sealing in water vapour than small glass vials. Furthermore, demineralisation of the residue with HCl/HF to isolate IOM could then be proceeded in the same vial without the need to transfer the residue. As such, for the extraction vials no burning could be undertaken. For the 10 ml vials, it was noted that heating at elevated temperatures deformed the glass and made them worse at sealing in volatiles during derivatisation. Also, it was noted that the glass was weakened by this process and the tubes would sometime crack during freezing if they had been heated beforehand. We have now included several sentences to clarify this in text (lines 229-230 and 233-235).

Were any steps taken to facilitate liberation of amino acids from the grains, e.g. crushing or sonication?

We noticed that for Ryugu particles when they were put into the water, they effectively broke up into powder themselves without the need for crushing as they were so fragile. Orgueil does the same, but Murchison for example, would need to be either crushed or powdered using a microtome. We have now included some text in the methods section to explain this (lines 239-240).

Thank you for the explanations and clarifications in the manuscript. Honest and very accurate description of sample treatment, contamination investigations, and cleaning procedures as well as procedural blanks results treatment is inevitable in this research, helps to strengthen the trustiness of the results as well as helps other scientists to perform analogous analyses. This is particularly important for the returned samples which are extremely precious.

2)

Can ~1 and ~2 mg samples be representative for drawing conclusions about the formation history of the organics in Ryugu?

We don't claim that the samples are representative of the entirety of Ryugu. What we say is that they are representative of the particles from which they came and as we have the modal mineralogy for these particles, we can discuss the organic matter's formation history in the context of the mineralogy. I don't think any study on Ryugu can claim to be representative of the entire asteroid, as only 5.4 g was returned.

Starting in line 149 you make a general conclusion saying: "As such, Ryugu should have experienced more intense aqueous alteration compared to Murchison (CM2), but less intense aqueous alteration compared with Orgueil (CI1)." This is based on the Beta-alanine/glycine ratios in the two particles while making a claim about the whole asteroid.

What about inhomogeneity of the distribution of organics? Clearly the particles you had are artificially defined. If you would have analysed a bigger chunk of Ryugu from the same site, it is possible that due to potential inhomogeneity of the distribution of organics you would have ended up with different relative abundances. Already particle A0106 analysed by Parker et al. (the LPSC conference website clearly states that the abstracts are peer-reviewed) comes from the same Chamber A and has a very different Beta-alanine/glycine ratio.

3)

The authors do not cite the first literature on Ryugu analysis by Parker et al. (LPSC, 2022), where amino acids and their concentration in 13.08 mg of A0106 Ryugu sample (much bigger piece of the asteroid) were reported.

We did not cite Parker et al., 2022 as this is a conference abstract and presumably thus not peer reviewed. We understand that this will likely be published as an article at some point and at that point we would be happy to reference it in a future article. We would also like to add that our group also reported our earlier findings in the Hayabusa2 symposium (2021), as did other groups, and we have not included any of those conference abstracts for the same reason.

Even though the A0106 (Parker et al.) and A0022 (present manuscript) samples come from the same Chamber A, i.e. the samples collected during the first touchdown sampling, their composition is different. Notably, in the present manuscript authors extensively discuss the ratio of β -alanine/glycine and draw implications about the extent of aqueous alteration that Ryugu would have experienced. For C0008 they report this ratio to be 0.57 and for A0022 1.04. Since these are between 0.26–0.43 in Murchison and 2.57–3.16 in Orgueil, they conclude that Ryugu must have experienced more intense aqueous alteration than Murchison and less intense than Orgueil. However, the A0106 sample analysed by Parker et al. has a β -alanine/glycine ratio of 7.14, i.e. significantly different even though this sample should be from the same collection site as A0022 from the present study. The authors argue that the β -alanine/glycine ratio is significantly different between A0022 and C0008, however, there is apparently greater difference between A0022 and A0106. The values of 2.54 and 3.12 for Orgueil in the present study are not far from the literature range (2.57–3.16), thence the sample treatment and derivatisation procedure should not have played a great role in affecting this ratio? Interestingly, N,N-dimethylglycine, the most abundant amino acid in the A0022 sample was not reported by Parker et al., possibly due to different sample extraction, derivatisation and analytical protocol.

Since the data of A0106 has not been published in a peer reviewed paper, we cannot evaluate why A0106 has a distinct β -alanine/glycine ratio at this moment. It is possible that some of the Chamber A particles, being on the surface of Ryugu, could be from exogenous micrometeorite impacts. Luckily, for our A0022 particle we have modal mineralogy data as we conducted a comprehensive geochemical analysis of our particles. As such we are able to confirm that it has a CI1-like composition and is thus most likely from Ryugu as it is similar in that respect to all of our 16 Ryugu particles from both touchdown sites that we analysed.

We agree that the procedure should not have played a major role in affecting the data. We agree that the HPLC procedure and the type of derivatisation we employed will mean that the

amino acids we can detect and their sensitivities will likely be different to the Parker study and that may explain this.

The LPSC conference website clearly states that the abstracts are peer-reviewed and hence Parker et al.'s data should be considered. Moynier et al. (Geochemical Perspectives Letters, 2022) also analyzed the calcium isotopic composition of A0106-A0107 particles.

4)

In addition to the hot water hydrolysis, further HCl hydrolysis of the samples could have proceeded during the derivatisation protocol. Did the authors consider potential ester hydrolysis of the derivatives during the HPLC analysis given an acidic mobile phase was employed?

We are not sure we completely understand. Are you saying that the amino acids derivatised as isopropyl esters could break down somewhat during the HPLC method as water reacts with the isopropyl esters? Or that the formic acid could form further esters from unreacted amino acids during the HPLC method? If so, then this shouldn't affect our results, because our standard was derivatised in the same way and run through the same method, so it would have been affected in the same way as the sample.

Ok, thank you.

5)

Authors argue in the Introduction section that so far only extra-terrestrial sources have been found to contain amino acids with an L-excess. This is, however, not a fair claim, since another source of amino acids could have been on the early Earth, however, the detection of an L-excess of these is obviously virtually impossible in the present-day biosphere, not meaning there was none.

We don't say that there were no potential environments on the Early Earth that could have generated L-excesses of amino acids, just that the only environments known to contain these are extraterrestrial ones. No one has so far proposed a plausible way to produce L-excesses of amino acids on the early Earth. As such, we don't think what we have said is wrong or misleading.

You can also look at this as follows. Where else than in the extra-terrestrial sources could you find L-excess in amino acids? On Earth. All amino acids are in huge L-excess on the present-day Earth. Hence, the statement as it is in the manuscript is not true.

6)

Furthermore, the authors argue that only simple amino acids can be produced in simulated interstellar ice experiments and more complex amino acids require aqueous processing. It is, however, ambiguous what the authors mean by "complex". At least 26 different amino/diamino acids were identified in simulated interstellar ices (water, methanol, ammonia) in the study by Meinert et al., ChemPlusChem, 2012. Moreover, the molecular inventory produced in the interstellar ices very much depends on the starting material and

hence it does not mean that if certain amino acids were yet not detected in the simulated ices, they cannot be formed there.

Meinert et al., 2012 does not quantify their amino acids, so it is hard to understand if like other studies (e.g. Modica et al., 2018 and Oba et al., 2016) many of the longer chain length or larger mass amino acids (more complex) are only present as small amounts compared to say glycine, alanine and B-alanine. No interstellar ice analogue study has so far demonstrated that they can produce amino acids at carbonaceous chondrite abundances. As such, aqueous alteration on meteorite parent bodies has been suggested (e.g. Modica et al., 2018, Martins et al., 2015, Glavin et al., 2010) to be responsible for increasing all the other amino acids relative to Glycine. As far as we know glycine is usually the most abundant amino acid in interstellar ice analogue studies and even B-alanine is low compared to it, being quite a bit lower than in most CM, let alone CI chondrites. The Paris meteorite is the closest in terms of the β -alanine/glycine among the CMs (e.g. Modica et al., 2018). As such, in terms of the experiments so far undertaken, we don't think it is misleading to say that aqueous alteration is likely required to explain the abundances of the amino acids in Ryugu, being that it is of a CI1-like composition (highly aqueously altered). We have now clarified meaning of simple and complex amino acids in text (Line: 56-58).

Modica et al., 2018, themselves cite Meinert et al.'s relative quantification (Table S1 in Meinert et al., 2012). Here, Beta-alanine is much less abundant than other C3 amino acids or even less abundant than norvaline (C5) or some unidentified C6 amino acids.

It should be stressed that interstellar ice simulation experiments are extremely valuable for understanding the mechanisms by which organics can be formed in space as well as what variety of species can be produced in space. However, they are generally performed only over a period of few hours up to several days, at pressures (and/or temperatures) higher than in space, they are limited in the variety of starting molecules that is predefined (often solely H₂O, NH₃, and CH₄) and the energy source is often limited to either a short wavelength range/monochromatic light or high energy electrons of single defined energy. Therefore, it is questionable whether the relative amino acid quantities from the ice simulation experiments are a reliable representation of the ones produced in real interstellar ices in space. As it comes to the absolute quantities, it is unsurprising that short ice simulation experiments cannot reproduce all the content in meteorites that was likely formed over much longer timescales.

It is still ambiguous how much is "low" molecular weight and how much is "high".

7)

Authors argue that many amino acids could be detected in one sample, but not the other (including protein amino acids which could be at least partly due to contamination). However, one has to keep in mind that the A0022 was only about half mass of the C0008 sample, with such small quantities and expectable sample inhomogeneity this is in my view not enough to draw any concrete conclusions about differences in formation history based on the differences in composition of these "single-shot" experiments from two different collection sites.

We do not use any of the low abundance amino acids to draw any formation history conclusions. We use the presence of DMG in A0022 and the β -alanine/glycine mostly. These amino acids are present at quite high abundances. We make mention of the differences in amino acid abundance between the two samples, but for the exact reason highlighted above by the reviewer we do not focus on the differences of the lower abundance amino acids. This again should alleviate any contamination concerns as well, because the main implications of the paper rest on some of the most abundant amino acids, which had some of the least contamination when looking at the blank traces in figure 1.

Explanation of procedural blanks analyses helps to eliminate doubts about potential contamination contributions to Gly, Beta-Ala, and DMG. However, my concerns about inhomogeneous distribution of organics within the asteroid remain. Your technique is extremely sensitive and you analysed very small pieces of two different sites of the asteroid. In case of DMG (highest abundance) it is a difference of 5.8 ng you detected in the 1mg grain of A0022 vs ~ 0 ng in 2 mg of the C0008 particle. These two small particles were artificially predefined and I am wondering about whether if you got instead a 10 mg particle encompassing grain A0022 (e.g. including particle A0106 analysed by Parker et al.), the relative abundances of amino acids you detected might be significantly different and hence also your hypotheses about their formation history.

In line 171 you make a claim about higher concentration of serine in A0022 compared to C0008 and link it to accretion of larger quantities of ice. Why the SD of serine is 12-times higher in A0022 as opposed to C0008 if the concentration is 4-times higher in A0022 as opposed to C0008?

You claim that aspartic acid was present at higher concentration in C0008 than A0022. Why the SD for aspartic acid differs so much in between the two particles?

Why larger Beta-alanine over glycine ratio is related to larger ice accretion in line 172, while in line 151 it is related to higher level of aqueous alteration?

In summary, the analysis of such small sample quantities seems very useful for a first-shot analysis to get an idea about what compounds are expected to be present in a new yet-uncharacterised sample, however, it does not seem the method of choice for concentration determination due to limited signal intensities in combination with potential contamination, which cannot be fully excluded. In addition, such small fragments cannot be a good representation of the inhomogeneous asteroid body and hence conclusions about the formation history of the amino acids in Ryugu based on analysing such small fragments are very premature. Finally, even when taking the two particles' amino acid composition as they are, the authors can only hypothesize about their origin, especially the authors in their interpretations rely on an ambiguous statement that only a limited selection of simple amino acids can be formed in simulated interstellar ices.

We have to disagree with the reviewer and we feel that the reasons we have mentioned above more than justify why we believe our methodology is appropriate and the resultant findings provide a novel and interesting contribution to the scientific literature.

I still consider the results of the analyses preliminary to draw any conclusions about the formations of amino acids in the sampled particles from Ryugu. This is demonstrated by the fact that the authors can only, based on the previous literature, speculate how the organics they found could have formed naming all possible scenarios (accretion of ices, Strecker synthesis, Michael addition, aqueous alteration,...). A similar discussion was already done by Nakamura et al., 2022, for C0008 particle (also scheme in Fig. 3 is based on Nakamura et al., 2022, however without a reference). This does not diminish in any way the beauty of the technique you developed for the analysis of very small sample quantities which you publish in ACS Earth Space Com. In order to be able to draw more evidence-based conclusions about the formation pathways of amino acids in Ryugu, I consider it to be important to analyze larger sample quantities so that the abundances of a large variety of amino acids are not just at/below the detection/quantification limits as well as to analyze more sample grains so that we get a more complete picture. Given many possibilities for the formation pathways, there is always several potential speculations we can come up with, however, having more data will help to narrow down all the hypotheses. I believe that your protocol with the analytical data presented in the present manuscript will be worth publishing in a more specialized journal.

Reviewer #2 (Remarks to the Author):

I would like to sincerely thank the authors for adding in so much text and figure modifications in response to my request. I think this manuscript looks much improved and I eagerly await the science that will follow after publication of these important results.

REVIEWER COMMENTS

Editor

While we agree with Reviewer 2 about the somewhat preliminary nature of your findings, we do find the results of interest and can offer to publish this manuscript once further minor revisions have been made. Nature Communications does not allow for the citation of any conference abstracts as these do not go through the same level of peer review as a published scientific article. However, you will need to carefully caveat your findings regarding the sensitivity of your small sample sizes to inhomogeneity in Ryugu and thus note that these interpretations are therefore somewhat preliminary and speculative – please do this in reasonable detail for our broad audience. You should also make clear the differences between short duration laboratory experiments and long term alteration in an asteroid cannot be expected to be completely relatable. You will also need to address Reviewer 2's remaining points regarding consistency.

We have now included in several areas of the text new discussion of the small sample size and the uncertainty relating to any conclusions about the global scale amino acid inventory of Ryugu. We have also made reference to the fact that it could be possible for ISM amino acids to represent those in meteorites, but that this not been achieved experimentally and that the current work on aqueously altered meteorites shows a relationship between B-alanine/glycine ratio and levels of aqueous alteration, which supports the idea that aqueous alteration affects the amino acids within carbonaceous chondrites that have experienced higher levels of aqueous alteration. We have also included below a detailed point by point response to reviewer 2's comments. Note that we have kept the previous revisions in yellow and added the most recent ones in green highlighting.

Reviewer #1 (Remarks to the Author):

Starting in line 149 you make a general conclusion saying: "As such, Ryugu should have experienced more intense aqueous alteration compared to Murchison (CM2), but less intense aqueous alteration compared with Orgueil (C11)." This is based on the Beta-alanine/glycine ratios in the two particles while making a claim about the whole asteroid. What about inhomogeneity of the distribution of organics? Clearly the particles you had are artificially defined. If you would have analysed a bigger chunk of Ryugu from the same site, it is possible that due to potential inhomogeneity of the distribution of organics you would have ended up with different relative abundances. Already particle A0106 analysed by Parker et al. (the LPSC conference website clearly states that the abstracts are peer-reviewed) comes from the same Chamber A and has a very different Beta-alanine/glycine ratio.

We do agree that the particles analysed by our study may not represent the entirety of Ryugu. As such, we have now updated the text to explain the sample size issue (see lines: 150-151 and 159). It may be possible that the organic matter could be distributed inhomogeneously over the scale of the particles that we analysed, but it is strange that two particles from distinct regions of Ryugu would both record much lower B-alanine/glycine ratios compared to that of Parker et al (see lines: 150-153 and 159-165). It could also be the case that Parker et al. included some very

anomalous particles with very high B-alanine concentrations that have increased the overall B-alanine/glycine ratio. Such particles could have been exogenous in origin, with distinct mineralogy and geochemistry. This is unlikely to be the case for our analyses as we have geochemical and mineralogical information for the particles we analysed.

The LPSC conference website clearly states that the abstracts are peer-reviewed and hence Parker et al.'s data should be considered. Moynier et al. (Geochemical Perspectives Letters, 2022) also analyzed the calcium isotopic composition of A0106-A0107 particles.

Moynier only measures the stable isotopes of ^{40}Ca and ^{44}Ca , not the ^{48}Ca used to understand the thermal processing history of solar system material. Therefore, this is not a relevant reference.

You can also look at this as follows. Where else than in the extra-terrestrial sources could you find L-excess in amino acids? On Earth. All amino acids are in huge L-excess on the presentday Earth. Hence, the statement as it is in the manuscript is not true.

We now understand what the Reviewer is saying. It was our intention to indicate that extraterrestrial sources would have most likely had L-excesses of amino acids before life originated on Earth and that no abiotic environment on the Early Earth is known to be able to create L-excesses of amino acids. In other words, we were focussing on the past before life originated, not the present where it is abundant on Earth. The sentence has now been updated to include abiotically synthesised amino acids to highlight this point (see line: 38).

Modica et al., 2018, themselves cite Meinert et al.'s relative quantification (Table S1 in Meinert et al., 2012). Here, Beta-alanine is much less abundant than other C3 amino acids or even less abundant than norvaline (C5) or some unidentified C6 amino acids.

If B-alanine is less abundant than alanine and some more complex amino acids in Meinert, then that means it is likely not present at carbonaceous chondrite abundances. This makes it more likely that aqueous alteration is important, especially as the B-alanine/glycine ratio has been shown to be indicative of the level of aqueous alteration in carbonaceous chondrites. Anyway, what we are trying to say is that lower molecular weight amino acids like glycine and say alanine and to a lesser extend B-alanine, tend to be more abundant than most of the higher molecular weight amino acids. This is true for the most part even if we look at the table S1 from Meinert, definitely for glycine and alanine. The problem with the relative abundance table is that they don't take into account the tendency of different molecules to fragment to different degrees on entry into the TOF-MS. It can also be the case, especially with TOF-MS that the RF values in the instrument, along with other settings, can affect the sensitivity of the instrument to different mass ranges. As such, I would be quite hesitant of comparing the relative intensities of different mass amino acids from their data and saying it is a good indication of actual relative abundances. In the case of Modica, they use GC-MS, which is a much more

simplistic analysis technique that does not induce as many ion flight paths and the same potential for mass dependant sensitivity issues, as TOF-MS.

It should be stressed that interstellar ice simulation experiments are extremely valuable for understanding the mechanisms by which organics can be formed in space as well as what variety of species can be produced in space. However, they are generally performed only over a period of few hours up to several days, at pressures (and/or temperatures) higher than in space, they are limited in the variety of starting molecules that is predefined (often solely H₂O, NH₃, and CH₄) and the energy source is often limited to either a short wavelength range/monochromatic light or high energy electrons of single defined energy. Therefore, it is questionable whether the relative amino acid quantities from the ice simulation experiments are a reliable representation of the ones produced in real interstellar ices in space. As it comes to the absolute quantities, it is unsurprising that short ice simulation experiments cannot reproduce all the content in meteorites that was likely formed over much longer timescales. It is still ambiguous how much is “low” molecular weight and how much is “high”.

We don't disagree that interstellar ice experiments are very important, it is clear that the Paris meteorite for example has a striking resemblance to some of the interstellar ice analogue residues in Modica et al., 2018. This is not surprising, since the Paris meteorite is the most aqueously unaltered CM2 that we know of. Hence if it accreted ISM or outer PSN-like ices with amino acids, these should somehow resemble the products of irradiated ice experiments. However, in the case of an incredibly aqueously altered asteroid like Ryugu, it appears from our results that asteroidal processes have altered the initially accreted amino acid abundances, if they were originally like those of interstellar ice analogue experiments. This is in agreement with the many studies that have investigated the amino acid inventories of carbonaceous chondrites that have experienced different levels of aqueous alteration. It is difficult for us to comment on the potential for different interstellar ice analogue experimental setups to produce Ryugu-like or CI-like amino acid abundances, as none have so far (see lines: 57-59). However, we have now included that the higher molecular weight amino acids are also produced in the irradiated ice experiments (see lines: 41-42) and mentioned that current interstellar ice analogue setups likely don't relate entirely to the actual real levels of amino acids produced in ISM and outer PSN environments (see lines: 168-170). In terms of low molecular weight and high molecular weight, they are relative terms and we use them with context in the text that makes it clear what we mean.

Explanation of procedural blanks analyses helps to eliminate doubts about potential contamination contributions to Gly, Beta-Ala, and DMG. However, my concerns about inhomogeneous distribution of organics within the asteroid remain. Your technique is extremely sensitive and you analysed very small pieces of two different sites of the asteroid. In case of DMG (highest abundance) it is a difference of 5.8 ng you detected in the 1mg grain of A0022 vs ~0 ng in 2 mg of the C0008 particle. These two small particles were artificially predefined and I am wondering about whether if you got instead a 10 mg particle encompassing grain A0022 (e.g. including particle A0106 analysed by Parker et al.), the relative abundances of amino acids you detected might be significantly different and hence also your hypotheses about their formation history.

Actually, we think the fact that we got such a small particle aliquot with known mineralogy and geochemistry is very useful, because it allowed us to compare the amino acid inventory and the possible influence of aqueous alteration, inferred from the mineralogy and geochemistry. If we had measured a bigger sample then maybe the relationship between the amino acids and the mineralogy and geochemistry would have been lost, because it may have represented an average of different matrix regions and their associated amino acid abundances. We think that if we look at the differences between the two particles, we measured from very distinct regions of Ryugu, then you can get something of an idea of the variation in Ryugu (see lines: 159-165). Of course, once larger sample size analyses are published, it will be interesting to revisit our more localised analyses to see where they fit in, in the range of amino acid abundance values for Ryugu.

In line 171 you make a claim about higher concentration of serine in A0022 compared to C0008 and link it to accretion of larger quantities of ice. Why the SD of serine is 12-times higher in A0022 as opposed to C0008 if the concentration is 4-times higher in A0022 as opposed to C0008?

You claim that aspartic acid was present at higher concentration in C0008 than A0022. Why the SD for aspartic acid differs so much in between the two particles?

The SD depend on the variation in the peak areas for these amino acids among the 3 runs for each sample. It appears that there was more variation in the peak area for A0022. This may relate to the fact that there could be a difference of the abundances or types of soluble molecules in these two samples and these can affect the spray stability during ionisation by H-ESI. Overall, though the R^2 values of the calibration curves for serine and aspartic acid from both samples were good (>0.99), so this doesn't affect the quality of our results.

Why larger Beta-alanine over glycine ratio is related to larger ice accretion in line 172, while in line 151 it is related to higher level of aqueous alteration?

In reality the ratio is related to the levels of aqueous alteration, it was related to the amount of ice accreted, because this defines the water to rock ratio, which in turn likely affects the level of aqueous alteration. In other words, more ice = more water, which = higher aqueous alteration. This has now been clarified in text (see lines: 184-185).

I still consider the results of the analyses preliminary to draw any conclusions about the formations of amino acids in the sampled particles from Ryugu. This is demonstrated by the fact that the authors can only, based on the previous literature, speculate how the organics they found could have formed naming all possible scenarios (accretion of ices, Strecker synthesis, Michael addition, aqueous alteration,...). A similar discussion was already done by Nakamura et al., 2022, for C0008 particle (also scheme in Fig. 3 is based on Nakamura et al., 2022, however without a reference). This does not diminish in any way the beauty of the technique you developed for the analysis of very small sample quantities which you publish in ACS Earth Space Com. In order to be able to draw more evidence-based conclusions about the formation pathways of amino acids in Ryugu, I consider it to

be important to analyze larger sample quantities so that the abundances of a large variety of amino acids are not just at/below the detection/quantification limits as well as to analyze more sample grains so that we get a more complete picture. Given many possibilities for the formation pathways, there is always several potential speculations we can come up with, however, having more data will help to narrow down all the hypotheses. I believe that your protocol with the analytical data presented in the present manuscript will be worth publishing in a more specialized journal.

We have to disagree with the reviewer, while the analytical results do have their limitations, due to the sample size constraints, the authors main aim was not to report on the bulk amino acid inventory of Ryugu. Instead, the authors relate localised amino acid abundances, from two particles from distant regions of Ryugu, to their mineralogy and geochemistry and draw conclusions based on the levels of aqueous alteration experienced and ice accreted. The levels of aqueous alteration and ice accreted are based on the mineralogy, geochemistry and the amino acid B-alanine/glycine ratio. Because we used these multiple lines of evidence, we feel that our conclusions are reasonable. Furthermore, while we do refer to the literature concerning the formation of the amino acids, much work has been done in this literature to highlight the different potential formation pathways of different types of amino acids. To put it simply, there isn't just one pathway that forms all amino acids. Moreover, we introduce a new pathway (the Eschweiler–Clarke reaction) that has not been reported before in an extraterrestrial context to form the DMG, which has also not been discussed much in organic cosmochemical publications. We do agree that the publications to come that will report on larger sample sizes, will be exciting for understanding the bulk inventory of Ryugu. However, these will still suffer from questions concerning how representative they of Ryugu as a whole. In essence what we have done here, to probe the local amino acid compositions and put them in context of the geology, is actually a lot less speculative than trying to generalise the global geology and amino acid inventory of Ryugu, which could be quite heterogeneous. As such, we feel our work is of great broad interest and relevance to the entire planetary science community and likely other closely related disciplines.

Reviewer #2 (Remarks to the Author):

No extra remarks.